# Network analysis of patterns and relevance of enteric pathogen co-infections among infants in a diarrhea-endemic setting

E. Ross Colgate[1,2�], Connor Klopfer[3☱], Dorothy M. Dickson [1,2], Benjamin Lee[1,4], Matthew J. Wargo[1,2], Ashraful Alam [5], Beth D. Kirkpatrick[1,2], Laurent Hébert-Dufresne [1,3,6]*

**1** Translational Global Infectious Disease Research Center, University of Vermont, Burlington, Vermont, United States of America, **2** Department of Microbiology and Molecular Genetics, University of Vermont, Burlington, Vermont, United States of America, **3** Vermont Complex Systems Center, University of Vermont, Burlington, Vermont, United States of America, **4** Department of Pediatrics, University of Vermont Larner College of Medicine, Burlington, Vermont, United States of America, **5** International Centre for Diarrhoeal Disease Research, Bangladesh, Dhaka, Bangladesh, **6** Department of Computer Science, University of Vermont, Burlington, Vermont, United States of America

☱ These authors contributed equally to this work.
* laurent.hebert-dufresne@uvm.edu

**Data Availability Statement:** There are no primary data in the paper and our software is available on

## Abstract

Despite significant progress in recent decades toward ameliorating the excess burden of diarrheal disease globally, childhood diarrhea remains a leading cause of morbidity and mortality in low-and-middle-income countries (LMICs). Recent large-scale studies of diarrhea etiology in these populations have revealed widespread co-infection with multiple enteric pathogens, in both acute and asymptomatic stool specimens. We applied methods from network science and ecology to better understand the underlying structure of enteric co-infection among infants in two large longitudinal birth cohorts in Bangladesh. We used a configuration model to establish distributions of expected random co-occurrence, based on individual pathogen prevalence alone, for every pathogen pair among 30 enteropathogens detected by qRT-PCR in both diarrheal and asymptomatic stool specimens. We found two pairs, Enterotoxigenic *E. coli* (ETEC) with Enteropathogenic *E. coli* (EPEC), and ETEC with *Campylobacter* spp., co-infected significantly more than expected at random (both pairs co-occurring almost 4 standard deviations above what one could expect due to chance alone). Furthermore, we found a general pattern that bacteria-bacteria pairs appear together more frequently than expected at random, while virus-bacteria pairs tend to appear less frequently than expected based on model predictions. Finally, infants co-infected with leading bacteria-bacteria pairs had more days of diarrhea in the first year of life compared to infants without co-infection (p-value <0.0001). Our methods and results help us understand the structure of enteric co-infection which can guide further work to identify and eliminate common sources of infection or determine biologic mechanisms that promote co-infection.

GitHub: https://github.com/connor-klopfer/copathogen-ensembles.

**Funding:** The authors acknowledge support from the National Institutes of Health 1P20 GM125498-01 Centers of Biomedical Research Excellence Award (E.R.C., C.K., D.M.D., B.L., M.J.W., B.D.K., & L.H.-D.). The funders had no role in study design, data collection and analysis, decision to publish, or preparation of the manuscript.

## Author summary

Since the turn of the 21st century, improved laboratory methods have helped us better understand the complex nature of infection with gut pathogens among infants and children in low-and-middle-income countries where diarrhea is still a leading cause of death and illness. We now understand that infants are often infected with more than one gut pathogen at the same time, whether or not they have diarrhea, but we understand very little about whether specific pathogens co-infect and how this is important to child health. In this study we identified two pathogen pairs appearing together much more frequently than expected at random in two cohorts in Bangladesh. Cases where these co-infections occur tend to have more days with diarrhea in the first year of life, which is meaningful to child health.

## Introduction

The persistent global health challenge of diarrhea-associated morbidity and mortality among young children in low-and-middle income countries (LMICs) is well documented [1–4]. In a concerted effort to eliminate the excess burden of childhood diarrheal disease, recent large-scale studies have employed advanced molecular diagnostics to better understand the etiology of diarrhea [5–7]. These studies clarified the highest burden diarrheal pathogens, but also provided two key insights critical to child health. First, diarrheal pathogens are highly prevalent not only during acute illness, but also in asymptomatic stools when children are "healthy" [8]. Second, enteric infections, whether diarrheal or asymptomatic, most often consist of simultaneous co-infection with multiple enteropathogens.

Childhood diarrhea not only remains the second leading cause of post-neonatal child mortality, but also causes a substantial burden of immediate and long-term diarrhea-related morbidity, including: malnutrition, impaired gut function, dysregulation of inflammatory and immune responses, and delayed or impaired cognitive development [9–12]. Importantly, recent studies have shown asymptomatic enteric infections, in the absence of overt diarrhea, carry similar or higher risk for the same adverse outcomes such as impaired growth and lower cognitive scores [13–18]. These findings are particularly impactful considering the prevalence of asymptomatic enteropathogen carriage: among more than 25,000 stools tested in the landmark GEMS and MAL-ED studies, enteric pathogens were detected in 72% and 65% of asymptomatic samples, respectively [19, 20]. An exclusive focus on curbing diarrheal disease may be insufficient to improve child health.

Understanding childhood enteric infection, both asymptomatic and diarrheal, becomes more complex with high rates of simultaneous co-infection with multiple enteric pathogens [5, 19, 21–24]. Nearly half of diarrheal specimens and one third of asymptomatic stools in GEMS and MAL-ED had two or more pathogens detected, with a range of up to 11 co-infecting agents [6, 19, 20]. Substantial effort has focused on assigning diarrhea attribution to a single pathogen in the instance of co-infection [5, 6, 12, 19]; however we now know that the mono-pathogen model of enteric disease does not represent the overall infection burden of children in LMICs, potentially resulting in significant misalignment of prevention and clinical management strategies to improve child health.

Given the ubiquity of co-infections, we need to classify co-infections based on their prevalence and consequences. That is, are some co-infections more highly associated with diarrhea or severe health outcomes? Should clinicians and public health experts be attempting to diagnose specific patterns of infection? To answer these questions, new modeling techniques are

necessary to articulate the structure of co-pathogen relationships, whether they are random, and how frequently they occur.

To move beyond the mono-pathogen model and capture the ecology of enteric infections, we present a network approach to understanding the underlying structure of co-pathogen infection and consequences for child health in two birth cohorts in Dhaka, Bangladesh. We applied a configuration model, established in network science, to determine patterns of second order co-infection in both asymptomatic and diarrheal stools, and importantly, we identified co-infections that appear either more or less frequently than expected at random, implying particular pathogen-pathogen or pathogen-host relationships that may be relevant to child health.

## Methods

### Study design and participants

Data for these analyses were collected from birth to 53 weeks in the MAL-ED Bangladesh and PROVIDE Study birth cohorts. The studies were conducted concurrently in a high-density area of Mirpur, Dhaka, Bangladesh between 2009—2014. Detailed methods for both studies are published [25–27]. No MAL-ED infants received rotavirus vaccine, while all infants in the PROVIDE sub-set received Rotarix vaccine at 10 and 17 weeks. Sub-sets of infants were selected based on complete pathogen data at all timepoints, available only for vaccinated infants in PROVIDE, as well as matching geographic and age ranges, with 210 infants included from MAL-ED and 270 from PROVIDE. Anonymized MAL-ED data were accessed through ClinEpiDB [28], and de-identified PROVIDE data were obtained from a study statistician.

### Procedures

All diarrheal stool specimens were collected through active biweekly surveillance. One diarrheal specimen was obtained for each reported episode, defined as three or more loose stools in 24 hours. Diarrheal episodes were separated by at least two (MAL-ED) or three (PROVIDE) diarrhea-free days. Total days of diarrhea were calculated from surveillance records. Asymptomatic stools were collected at scheduled study visits: monthly in MAL-ED, and in PROVIDE at enrollment ($<$ 2 weeks post-partum) and 6, 10, 14, 18, and 24 weeks.

Simultaneous detection of 30 enteropathogens in each stool was done by quantitative polymerase chain reaction (qPCR) TaqMan Array Card assay (TAC). Detailed extraction and assay methods have previously been described [6, 29, 30]. To capture both sub-clinical and diarrhea-associated infections, a positive TAC result was considered any cycle threshold value less than 35. Pan-genus gene targets were used for *Campylobacter* spp. and *Cryptosporidium* spp., while species-level targets were used for other pathogens. Pathogenic *E. coli* spp. were separated into pathotypes using derivations based on nine gene targets, as previously described [6], except in this analysis all strains of Enterotoxigenic *E. coli* (including heat-labile and heat-stable toxins) were defined as ETEC, and typical Enteropathogenic *E. coli* was labeled EPEC. See Table 1 for all pathogens included in the analysis.

### Model

Our modeling approach was selected to maintain structural information about the dataset, specifically the degree distribution of both pathogen richness per stool and individual pathogen prevalence, with a null hypothesis that pairwise pathogen co-infection is due to individual

**Table 1. Pathogen targets evaluated by TaqMan Array Card qPCR.** For gene targets and pathogen derivations from multiple targets, see previous publication [6].

| Bacteria | Viruses | Helminth | Protozoa | Fungi |
|---|---|---|---|---|
| *Aeromonas* | Adenovirus 40/41 | *Ancylostoma duodenale* | *Cryptosporidium* | *Encephalitozoon intestinalis* |
| *Bacteroides fragilis* | Astrovirus | *Ascaris lumbricoides* | *Entamoeba histolytica* | *Enterocytozoon bieneusi* |
| *Campylobacter* spp. | Norovirus GI | *Cyclospora cayetanensis* | | |
| *Clostridium difficile* | Norovirus GII | *Cystoisospora belli* | | |
| EAEC | Rotavirus | *Necator americanus* | | |
| EPEC | Sapovirus | *Strongyloides stercoralis* | | |
| ETEC | | *Trichuris trichiura* | | |
| *Helicobacter pyroli* | | | | |
| *M. tuberculosis* | | | | |
| *Salmonella* | | | | |
| *Shigella*/EIEC | | | | |
| STEC | | | | |
| *Vibrio cholerae* | | | | |

pathogen prevalence independent of presence of other pathogens being present, therefore observed co-infection should be captured by a distribution of random co-occurrence. Our underlying assumption was that the structure of co-pathogen infection differs between asymptomatic infections versus diarrhea-associated infections.

To test this hypothesis, stool samples were partitioned by study of origin, either MAL-ED or PROVIDE, and type of stool, asymptomatic or diarrheal, to generate bipartite network graphs representing the relationship between stool samples and enteric pathogens. A presence-absence matrix for each partition was represented by a bipartite network, $n$ by $m$ matrix $X$, where $n$ was the number of stool samples for the given stool type in each study and $m$ was the number of possible pathogens per stool. An edge was drawn from stool sample $i$ to pathogen $j$ if that pathogen was observed in that stool sample, such that $X_{ij} = 1$.

We selected a conservative fixed degree sequence model (FDSM), to minimize Type 1 error in generating an ensemble of null communities [31, 32]. Observed bipartite networks were randomized, where an edge $X_{ij}$ was removed at random and redrawn from node i to a node sampled from the j population. Null communities were generated by randomly rewiring each network 15,000 times, resulting in a probability <.01% of missing an edge for re-wiring. This fixed degree randomization, preserving pathogen richness for each stool and pathogen prevalence, was repeated with a new copy of the presence-absence matrix to produce an ensemble of 10,000 null communities in the model.

Binomial distributions of expected co-occurrence for each pathogen pair were then derived from the population of $10^4$ randomizations of the original bipartite graph by generating one mode projections of group i with $XX^T$ and group j with $X^TX$. This yielded a p-value for each edge, representing $P(x < X)$, or the percentage of the null ensemble $E[X]$ that was less than the observed number of co-occurrences in the dataset. Significant pairs were considered those whose observed co-occurrence fell outside the 95% credible interval of the binomial distribution.

Pathogen pairs occurring >10 times were ranked according to the distance between the pair's $P(x < X)$ and the mean of the null ensemble, divided by the maximum rank value in each study, resulting in a rank in the range [0, 1]. Ranks for each pair were averaged between the studies to obtain an overall ranking per pathogen pair.

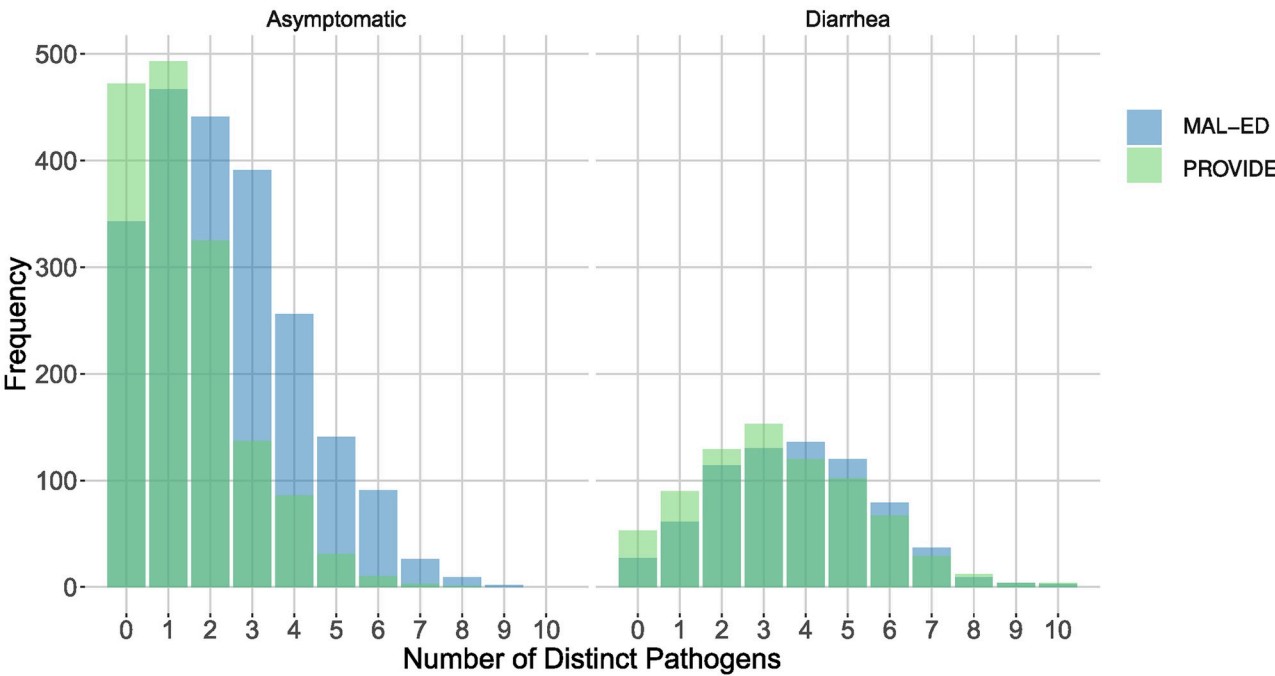

**Fig 1. Histograms of number of distinct pathogens per observation in both studies.** As in most of our results, we keep the MAL-ED and PROVIDE studies separate and split the observations into asymptomatic and diarrheal stool categories. In all cases, most observations contain significant co-infections.

## Results

### Descriptive results

We evaluated co-infections among 210 infants in the MAL-ED Bangladesh Study contributing 2,167 asymptomatic surveillance stools and 717 diarrheal stools, and 1,558 asymptomatic and 763 diarrheal stools from 270 infants in the PROVIDE Study. Infants were exposed to a diversity of enteropathogens in the first year of life with a mean of 11.32 (SD 1.71) distinct pathogens detected per child in MAL-ED and 8.1 (3.02) in PROVIDE. As shown in Fig 1, simultaneous co-infection with two or more pathogens was detected in 62.6% and 38.1% of asymptomatic stools and 87.8% and 81.6% of diarrheal stools in MAL-ED and PROVIDE respectively. The mean number of co-pathogens per diarrheal stool was 3.74 (1.92) in MAL-ED and 3.37 (2.02) in PROVIDE, and 2.34 (1.77) and 1.37 (1.34) pathogens per asymptomatic stool, respectively. Individual pathogen prevalence, held constant in the configuration models, for each pathogen appearing in the top ranked co-infection pairs are shown in S1 Fig.

### Configuration model results

Separate configuration models were developed for each study to identify co-pathogen pairs appearing either more or less frequently than expected against the null model in both asymptomatic and diarrheal stools. Figs 2 and 3 show the top 12 ranked co-pathogen pairs per stool type. Among the 12 highest ranked pairs in asymptomatic stools, bacteria-bacteria pairs accounted for the majority of co-infections, 7 of 12 pairs (58%). The remaining five top ranked asymptomatic pairs were bacteria-virus co-infections, each with rotavirus, norovirus GII or adenovirus 40/41 co-infecting with a bacterial pathogen. Pathogenic *Escherichia coli* strains

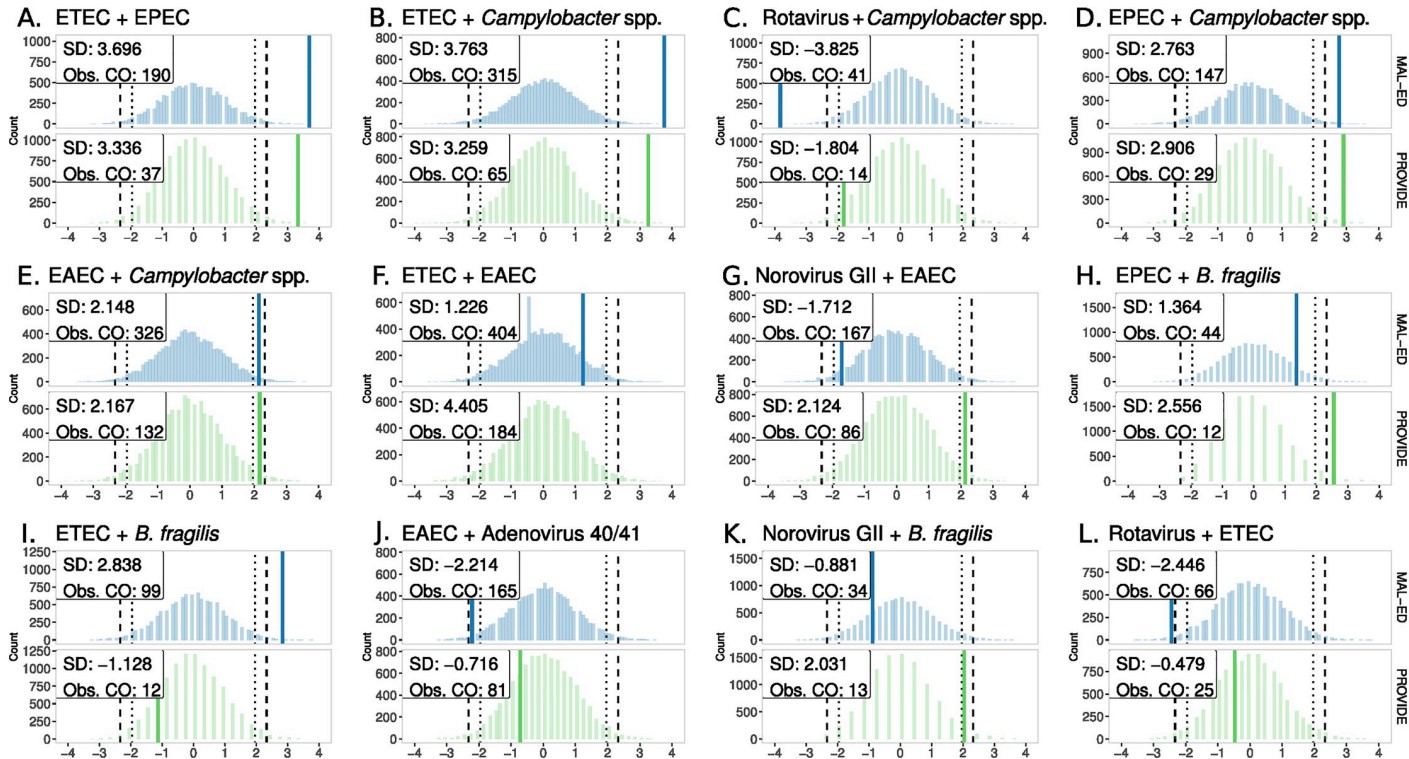

**Fig 2. Top 12 ranked pairwise co-infections in asymptomatic stools.** Background distributions for each study, blue for MAL-ED and green for PROVIDE, depict the null distribution of expected co-occurrence based on the configuration model for each pathogen pair. Solid colored vertical lines show the observed number of co-occurrences (Obs. CO) standardized by distance from the mean (SD). The solid colored vertical line is not visible for pairs where Obs. CO lies outside of the -4SD to 4SD interval. Dotted vertical lines represent the 95% credible interval (CI) for the expected number of co-occurrences based on the primary null configuration models. Hashed vertical lines show the 99% CI. Pairs are shown from highest ranking in the top left corner to the lowest in the bottom right of the panel. Ranking for each pathogen pair was calculated as the distance (Z score) of the observed absolute number of co-infections from the null mean in units of standard deviation. The study-specific ranks were then averaged to obtain an overall ranking per pathogen pair. ETEC = *Enterotoxigenic E. coli*. EPEC = *Enteropathogenic E. coli*. EAEC = *Enteroaggregative E. coli*. spp. = species. B. fragilis = *Bacteroides fragilis*.

were predominant in asymptomatic co-infections; either Enterotoxigenic *E. coli* (ETEC), Enteropathogenic *E. coli* (EPEC), or Enteroaggregative *E. coli* (EAEC) appeared in 10 of 12 (83%) highly ranked pairs. The second most common pathogen in top ranked asymptomatic pairs was *Campylobacter* spp. (4/12, 33%). Finally, Enterotoxigenic *Bacteroides fragilis* (ETBF) was found only in top pairs in asymptomatic stools, not in diarrheal stools, and co-occurred with EPEC, ETEC and norovirus GII.

Top pairs in diarrheal stools were balanced between bacteria-bacteria and bacteria-virus combinations with five pairs each, while only two pathogen sets of other etiologies ranked in the top 12: EPEC paired with the intestinal parasite *Cryptosporidium* spp., and the virus-virus pair of norovirus GII and astrovirus. Pathogenic *E. coli* pathotypes were again dominant in pathogen pairs in diarrhea as 9 of 12 (75%) top ranked pairs had at least one strain of *E. coli*. In diarrhea, all top bacteria-bacteria pairs were observed above the 95% credible interval (CI) of the corresponding null distributions, indicating higher-than-expected (HTE) co-occurrence compared to random co-infection. Three pathogens appeared uniquely in pairs found in diarrheal stools compared to asymptomatic: *Cryptosporidium* spp., *Shigella* spp. and astrovirus. Notably, three of the top 12 diarrheal pairs included *Shigella* spp., and when paired with *Campylobacter* spp. or ETEC, *Shigella* spp. co-infections occurred above the 99% CI in both MAL-ED (3.17 standard deviation (SD) and 2.70SD) and PROVIDE (2.60SD and 2.93SD), as

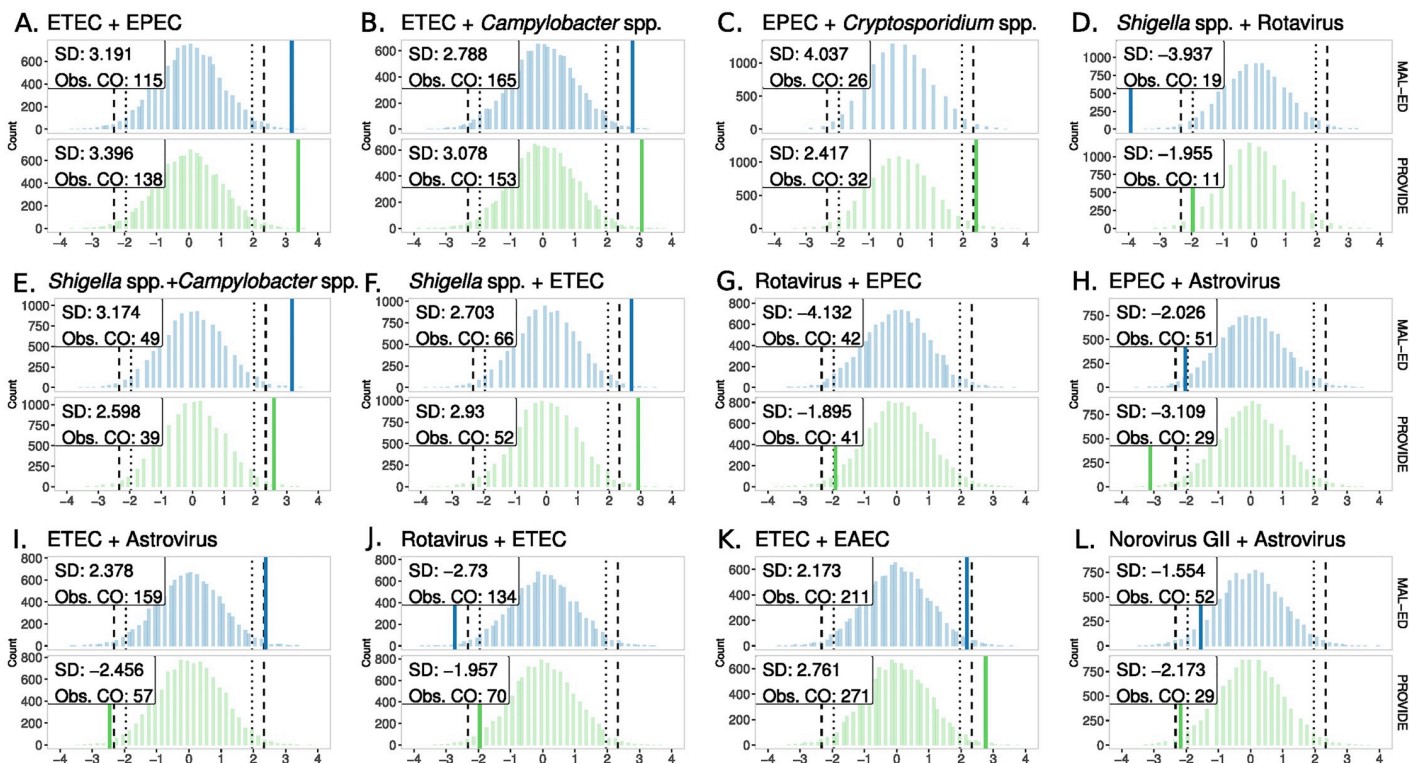

**Fig 3. Top 12 ranked pairwise co-infections in diarrheal stools.** Background distributions for each study, blue for MAL-ED and green for PROVIDE, depict the null distribution of expected co-occurrence based on the configuration model for each pathogen pair. Solid colored vertical lines show the observed number of co-occurrences (Obs. CO) standardized by distance from the mean (SD). The solid colored vertical line is not visible for pairs where Obs. CO lies outside of the -4SD to 4SD interval. Dotted vertical lines represent the 95% credible interval (CI) for the expected number of co-occurrences based on the primary null configuration models. Hashed vertical lines show the 99% CI. Pairs are shown from highest ranking in the top left corner to the lowest in the bottom right of the panel. Ranking for each pathogen pair was calculated as the distance (Z score) of the observed number of co-infections from the null mean in units of standard deviation. The study-specific ranks were then averaged to obtain an overall ranking per pathogen pair. ETEC = *Enterotoxigenic E. coli*. EPEC = *Enteropathogenic E. coli*. EAEC = *Enteroaggregative E. coli*. spp. = species.

opposed to *Shigella* spp. paired with rotavirus which occurred < 95% CI in both studies (–3.94SD and –1.96SD respectively), significantly less-than-expected (LTE) at random.

Four co-pathogen pairs appeared in the top 12 in both asymptomatic and diarrheal stools, each pair consisting of Enterotoxigenic *E. coli* (ETEC) with either Enteropathogenic *E. coli* (EPEC), *Campylobacter* spp., Enteroaggregative *E. coli* (EAEC), or rotavirus. Comparing the observed co-occurrence of these four pairs to their null distributions shows a pattern that applies widely across top-ranked co-infections: bacteria-bacteria pairs tend to appear at higher-than-expected (HTE) frequency compared to the null, whereas bacteria-virus pairs often appear less-than-expected (LTE) at random, see Table 2. Following the bacteria-bacteria pattern, in both MAL-ED and PROVIDE, EPEC + *Cryptosporidium spp.* were observed together HTE (> 99% CI) while norovirus GII and astrovirus co-occurred < 95% CI in PROVIDE (only), similar to the bacteria-virus pattern.

We examined concordance between the MAL-ED and PROVIDE studies in top pathogen pairs to distinguish cohort-specific effects from generalizable findings. Looking at both stool types, seven pairs were observed above the 95% CI in both MAL-ED and PROVIDE, all of which were bacteria-bacteria pairs consisting of combinations of ETEC, EPEC, EAEC, *Campylobacter* spp. and *Shigella* spp., see Figs 2 and 3. Notably three of the four HTE pairs in both studies in asymptomatic stools were *E. coli* pathotypes paired with *Campylobacter* spp. (the

**Table 2. Standard Deviations for pathogen pairs ranked in top 12 for both asymptomatic and diarrheal stools, by study.** Cells are colored based on significance, i.e., where the observed deviation falls in the credible interval based on the null model. ETEC = Enterotoxigenic *E. coli*. EPEC = Enteropathogenic *E. coli*. EAEC = Enteroaggregative *E. coli*. spp. = species.

| Pathogen Pair | Asymptomatic | | Diarrheal | |
|---|---|---|---|---|
| | **MAL-ED** | **PROVIDE** | **MAL-ED** | **PROVIDE** |
| ETEC + EPEC | 3.70 | 3.34 | 3.19 | 3.40 |
| ETEC + *Campylobacter* spp. | 3.76 | 3.26 | 2.79 | 3.08 |
| ETEC + EAEC | 1.23 | 4.41 | 2.17 | 2.76 |
| ETEC + Rotavirus | -2.45 | -0.48 | -2.73 | -1.96 |

■ SD < -99%  ■ -99% < SD < -95%  ■ -95% < SD < 95%  ■ 95% < SD < 99%  ■ 95% < SD < 99%

fourth was ETEC+EPEC), while pairs including *Shigella* spp. appeared HTE in both studies only in diarrheal stools (*Shigella* spp. + *Campylobacter* spp. and *Shigella* spp. + ETEC). Two co-pathogen pairs occurred HTE (> 99% CI) in both stool types and both studies: ETEC + EPEC and ETEC + *Campylobacter* spp. By contrast, no pathogen pair was observed below the 95% CI (LTE) in both MAL-ED and PROVIDE in asymptomatic stools, and all three concordant LTE co-occurrences in diarrheal stools were bacteria-virus pairs consisting of rotavirus paired with ETEC or *Shigella* spp., and EPEC with astrovirus.

While rotavirus co-infections with ETEC or *Shigella* spp. were consistently LTE in diarrheal stools in both studies, other rotavirus-inclusive pairs diverged between the studies. Across all pathogen pairs occurring more than 10 times in the dataset, five rotavirus-inclusive pairs appeared in both asymptomatic and diarrheal stools: rotavirus with *Campylobacter* spp., ETEC, EAEC, norovirus GII and astrovirus, see Fig 4. With the exception of rotavirus + ETEC in diarrhea, all significantly LTE (< 95% CI) rotavirus co-infections occurred only in MAL-ED, whereas in PROVIDE these pairs were observed within the expected 95% range per the null distributions in both asymptomatic and diarrheal stools.

See S2 and S3 Figs for the null distribution figures for all co-pathogen pairs occurring > 10 times in the dataset. S2 Fig highlights internal model validation, as the majority of possible co-infections occurred within the expected range in asymptomatic stools (26 of 42 pairs, 62%) and

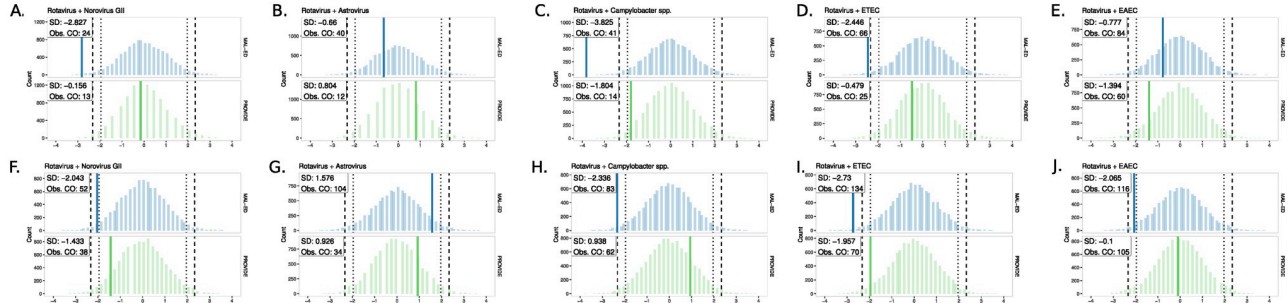

**Fig 4. Pairs with rotavirus have discordant significance between the studies.** Background distributions for each study, blue for MAL-ED and green for PROVIDE, depict the null distribution of expected co-occurrence of pairs including rotavirus based on the configuration model for each pathogen pair. For each vertical pair of figures, e.g. panels A and F, asymptomatic tools are shown on top and diarrheal stools on the bottom. As before, solid colored vertical lines show the observed number of co-occurrences standardized by distance from the mean. Dotted vertical lines represent the 95% credible interval (CI) for the expected number of co-occurrences based on the primary null configuration models. Hashed vertical lines show the 99% CI. Pairs shown include all co-occurrences with rotavirus among pairs observed > 10 times in the dataset. ETEC = *Enterotoxigenic E. coli*. EPEC = *Enteropathogenic E. coli*. EAEC = *Enteroaggregative E. coli*. spp. = species.

nearly half were observed as expected at random in diarrhea (17 of 43 pairs, 40%): the model accurately predicted the range of expected frequencies for the majority of pathogen pairs.

## Diarrheal burden of co-pathogen infection

The overall frequency of top-ranked pathogen pairs per stool type ranged from 0.8%—18.6% in asymptomatic stools and 1.4%—35.5% in diarrheal stools (S1 and S2 Tables) This modest prevalence of the most significant pairs may mask important effects of these co-infections on child health. For each top-ranked pair, the proportion of stools that were diarrheal exceeded the overall proportion of diarrhea in the analysis: whereas 1,480 stools out of 5,205 total samples (28%) were diarrheal, the proportion of stools that were diarrheal among top pathogen pairs ranged from 33%—79%. Fig 5 shows the risk of diarrhea among all stools containing the top pathogen pairs shown in Figs 2 and 3. Confidence intervals (CI) for relative risk can be generated by taking the lowest CI bound for risk of co-infection in diarrheal stools over the highest CI bound for the risk of co-infection in asymptomatic stools (both calculated using Wilson score interval), and vice versa to get an upper bound on relative risk. Doing so, we find when highest relative risk of a given stool being diarrheal, as opposed to asymptomatic, was found where *Cryptosporidium* spp. co-infected with EPEC, with 5.94 (95% CI [3.10, 11.40]) times higher risk of finding this co-infection in diarrhea. Likewise, the relative risk of diarrhea was higher with *Shigella* spp. + either ETEC or *Campylobacter* spp. co-infection (5.09 [3.30, 7.86] and 3.94 [2.46, 6.28] times higher risk). *Campylobacter* spp. paired with *E. coli* spp. also had an increased diarrheal risk of 1.75 [1.44, 2.11], 2.06 [1.68, 2.52], and 2.24 [1.66, 3.01] times greater than asymptomatic stools among co-infections with EAEC, ETEC and EPEC, respectively. For bacteria-virus pairs, the highest risk of diarrhea was found in rotavirus-associated pairings with *Shigella* spp. (9.22 [3.28, 25.9] times greater risk of diarrhea), *Campylobacter* spp. (6.48 [4.25, 9.88]), ETEC (5.51 [3.94, 7.71]), and EPEC (5.23 [3.10, 8.83]). In contrast, the lowest diarrheal risk was among pairs that included Enterotoxigenic *B. fragilis*, suggesting minimal diarrheal risk with carriage of this enteropathogen, regardless of co-pathogen species.

Finally, our results on co-occurrences can be summarized as networks of co-infection patterns that deviate from expectations drawn from the Configuration Model, see Fig 6. Therein, loops of overrepresented, underrepresented, and mixed co-occurrences suggest that future higher-order analysis of co-occurrences patterns beyond pairs of pathogens will be necessary.

## Discussion

Despite substantial data demonstrating a heavy burden of both symptomatic and subclinical enteric co-infection in infants in low-and-middle-income countries (LMICs), patterns of pairwise enteric co-infections have remained largely uncharacterized. For example, the data used here show that most diarrheal stools have more than three pathogens and a third of asymptomatic stools have at least two pathogens. Our goal was therefore to determine which specific pathogen pairs appear more or less than expected at random. Our study leads to a better understanding of the underlying structure of enteric infection in a population of Bangladeshi infants, which may have implications for child health, as described below.

We chose a configuration model approach specifically to establish null distributions of expected random co-occurrence based on a given number of pathogens per stool and fixed prevalence of each pathogen. With these parameters, two pathogen pairs, Enterotoxigenic *E. coli* (ETEC) with Enteropathogenic *E. coli* (EPEC) and ETEC with *Campylobacter* spp., were observed above the 99% credible interval (CI) in both studies and in both asymptomatic and diarrheal stools: it is very unlikely these pathogens co-occur in a random or study-specific

## A. Bacteria with non-virus

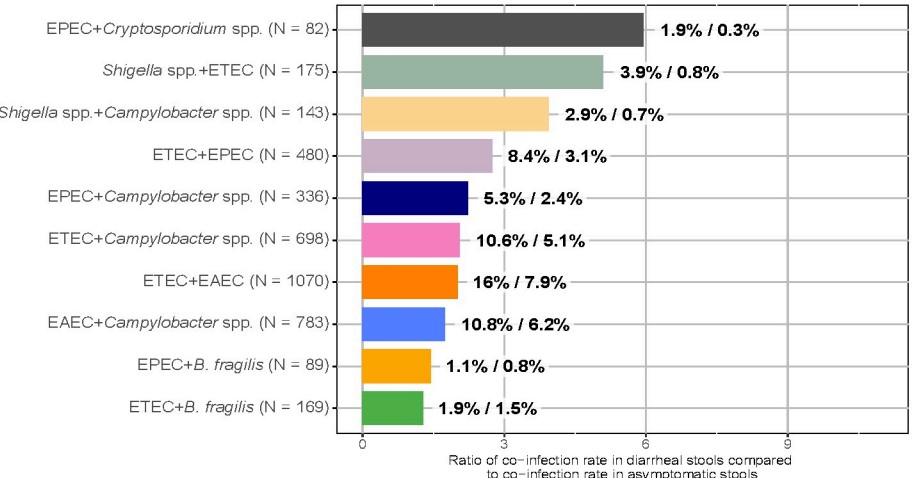

## B. Virus with virus or bacteria

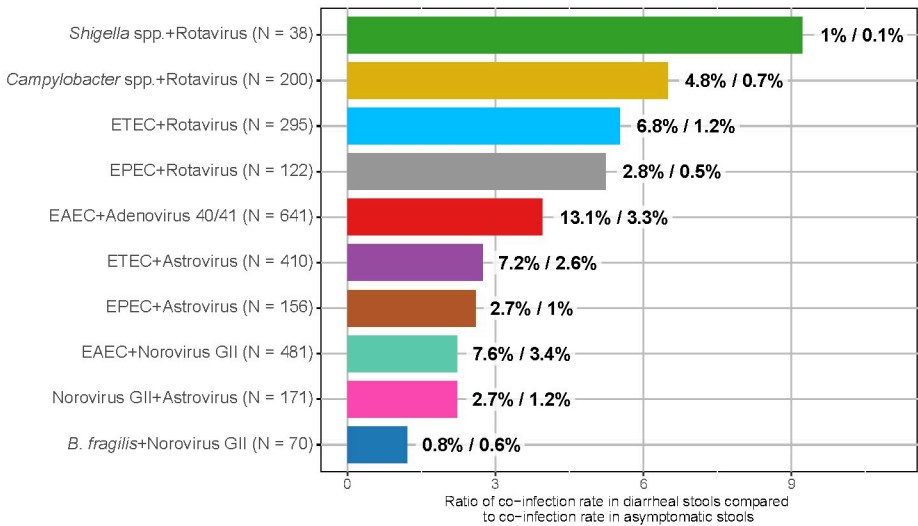

**Fig 5. Risk of co-infection in diarrhea compared to asymptomatic stools in top pairs by etiology.** Panel A shows pairs excluding viruses. Panel B shows bacteria-virus pairs. Bar charts show the risk of diarrhea among all stools with each pathogen pair. Total number of stools with each pathogen pair is shown in the legend ($N = X$), while the percent of co-infection in diarrheal stools over the percent of co-infection in asymptomatic stools is shown next to each bar. ETEC = *Enterotoxigenic E. coli*. EPEC = *Enteropathogenic E. coli*. EAEC = *Enteroaggregative E. coli*. spp. = species. *B. fragilis = Bacteroides fragilis*.

manner. Importantly, infants with these concurrent co-infections had as much or more diarrhea on average than the overall cohort in the first year of life.

Our results raise questions about pathogen interactions, including possible shared transmission pathways, similar environmental responses, persistent infections, or synergistic relationships that promote co-infection. As a primary example, many enteric infections are known to be mediated by seasonal variations affecting the fecal-oral transmission route through changes in the use of ground and well water around the wet season [33]. We would

## A. Diarrheal in MAL-ED

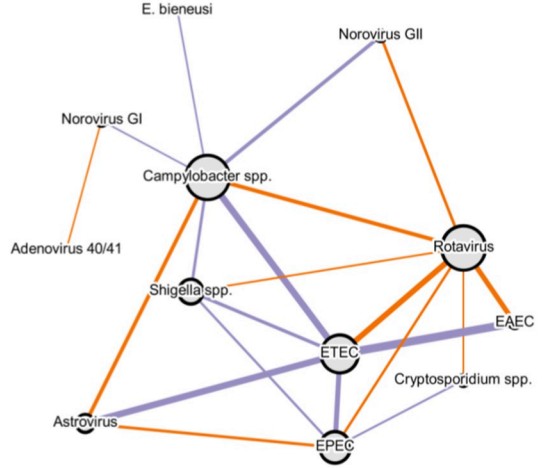

## B. Asymptomatic in PROVIDE

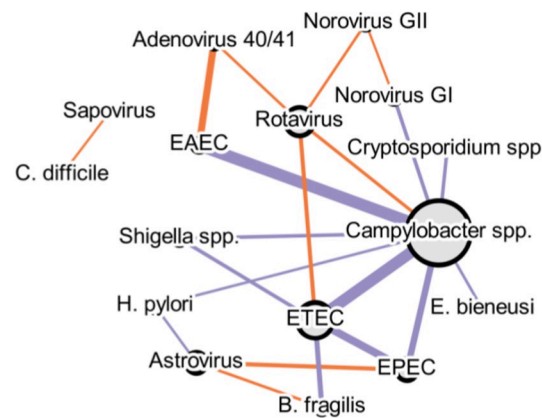

## C. Diarrheal in PROVIDE

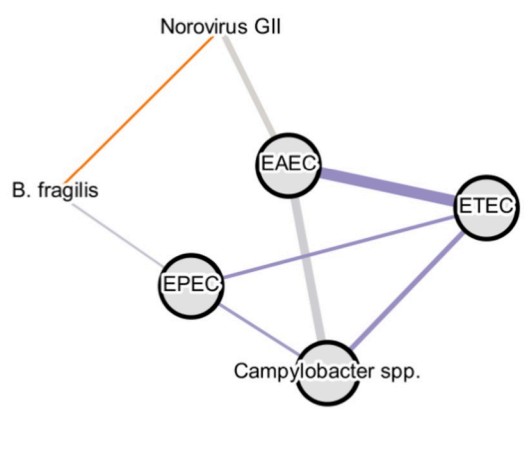

## D. Asymptomatic in PROVIDE

**Fig 6. Summary of the major pairwise co-occurrence signal identified in this paper.** Pairs of pathogens co-occurring more (less) than expected at random are shown in purple (orange). Stools are separated by type and study.

expect co-infections to align with this seasonal pattern, which was not tested here. As a second example, separate murine models of ETEC and *Campylobacter* showed enhanced bacterial growth in the setting of host zinc deficiency [34, 35], so one might expect higher co-infection in zinc deficient hosts through nutrition- or environment-mediated co-infection. These pathogens may share a response to prevalent zinc deficiency in infants in LMICs, contributing to higher-than-expected co-infection rates. Such environmental factors could be identified using nutritional and environmental covariates and models such as joint-species distribution models from ecology [36]. As a third example, a novel hypothesis based on *Campylobacter* human challenge models demonstrating recrudescence [37] is that two bacteria prone to persistent infection may establish independent niches in the intestine leading to longitudinal co-

infection. Relevant to a high burden asymptomatic pair identified in our analysis, multiple rat models demonstrated a dose-response synergistic effect of co-infection with *E. coli* and *B. fragilis* ranging from intra-abdominal abscess formation to lethality depending on the initial inocula of bacteria and when compared to mono-infections with either pathogen. One explanation for these observations was that an extracellular metabolite from *B. fragilis* nourishes *E. coli*, enhancing both the abundance and virulence of the co-infection [38–40].

We demonstrated increased relative risk of top pathogen pairs occurring in diarrheal, as opposed to asymptomatic, stools. These effects are important but are also confounded with many variables other than co-infections, perhaps most importantly with age. Beyond age, host susceptibility, mucosal immunity, level of exposures, nutritional status, and many other factors are known to contribute to asymptomatic enteric infections [8].

Vaccines are another relevant exogenous factor, as demonstrated by our finding that the vast majority of rotavirus co-infections occurred only in MAL-ED. This is notable as no children in MAL-ED were immunized against rotavirus, while all PROVIDE children in this analysis received two doses of Rotarix vaccine. Accordingly, and providing model validation, rotavirus co-infections appeared mostly within the expected range for PROVIDE infants: rotavirus was not a primary driver of infection. By contrast, in unvaccinated MAL-ED infants, the majority of rotavirus-associated co-infections appeared less frequently than expected at random, implying rotavirus dominated the infection landscape in this population acting more as a single antigen as opposed to occurring in co-infections.

Time remains an important aspect missing from our primary analysis. We hypothesize the age of the child at time of pathogen(s) exposure, the longitudinal sequence of mono- and co-infections, seasonality, and timing of prevention and treatment interventions may affect networks of co-infection over time. The main analysis here pools samples collected over the first 53 weeks of life; however future analyses will incorporate time as a key feature of enteric co-infection. We tested the possibility of including time in our model by partitioning the data according to an appropriate time window based on child's age before generating the null ensembles. The result may be seen in S4 Fig. Similar to Fig 6, the results are presented by study and stool type, with the extra partition of a time point, either 1, 3 or 6 months of age. We see the dynamics of some co-infections change when including time, for example the strength of the relationship between ETEC and EPEC weakens with age-based partitioning. For our purposes, it is difficult to draw conclusions from these partitions, because sub-setting the data this way reduces the sample size dramatically. Future work would need to overcome this limitation to avoid skewed distributions.

Partitioning the data set can be extended to environmental factors such as location, socio-economic status, nutrition, or even endogenous risk factors of disease. For the current analysis only the Bangladesh data from MAL-ED was used in order to compare the results from two different studies in the same geographic area. An obvious next step is to expand our analysis to partition data from all MAL-ED study sites based on location.

Finally, future work on co-infections will need to address additional limitations of our work. Here, each pairwise co-occurrence was considered independent, regardless of additional co-infecting pathogens, which is insufficient to describe higher-order co-infections. Different methods with some relaxed assumptions are required to accommodate greater complexity, S5 Fig. Interpretation of our results is also limited to significant co-occurrences, which may be distinct from pathogen interactions [41]. Possible interactions, as hypothesized here, must be explored separately and potentially by looking at patterns beyond pairwise co-infections [42]. Importantly, our analyses did not include quantitative aspects of the TaqMan polymerase chain reaction data for individual pathogens: all pathogens were defined as present or absent based on the generic positivity range of ($0 < CT < 35$). Assigning biological relevance to CT

values, particularly in the context of co-infection, remains an open question in the enterics field, and future modeling efforts incorporating quantitative CT data will require clear articulation of assumptions distinguishing active infections versus subclinical carriage, especially in asymptomatic stools. Lastly, there is a dire need for studies exploring the impact of significant co-infections on diarrheal burden in the first years of life without having to rely on ecological approaches.

In summary, we applied an established method from network science to better understand the structure and consequences of enteric co-infection in infants. We identified significant pairs of pathogens which could be prioritized for consideration in developing and deploying interventions to ameliorate the excess burden of enteric infection in this population. Further research to understand the mechanisms promoting these co-infections will help guide intervention strategies.

## Supporting information

**S1 Fig. Pathogen prevalence for individual pathogens found in top pathogen pairs in asymptomatic or diarrheal stools, ranked by their deviance from the null ensemble.** (PNG)

**S2 Fig. All null ensemble distributions from the configuration model of asymptomatic stools for pathogen pairs appearing $> 10$ times.** Includes top 10% of all pathogen pairs, ranked by the percentile's distance from 0.5. With 435 possible combinations, 42 pairs make up the top 10% possible. (PDF)

**S3 Fig. All null ensemble distributions from the configuration model of diarrheal stools for pathogen pairs appearing $> 10$ times.** Includes top 10% of all pathogen pairs, ranked by the percentile's distance from 0.5. With 520 possible combinations, 52 pairs make up the top 10% possible. (PDF)

**S4 Fig. Time partitioned networks generated by partitioning stool samples based on age at collection.** Each row represents a study and stool type, and each column represents a time window, either 1, 3 or 6 months. Networks were generated using the same methods as described above, using subsets of the original data based on the time of collection. Connections between pathogens are shown if they were observed to happen greater than ten times, and the saturation of color represents the distance from the null distribution average, where purple (orange) means the co-occurrences occurred higher (lower) than the null distribution, gray indicates the co-occurrences appeared at a rate similar to the average in the null distribution. (PNG)

**S5 Fig. Comparison with HOLMES, a networked $\chi^2$ test [42].** HOLMES is a generalization of $\chi^2$ test which compares pairs and higher-order groups for significant interactions in presence/absence data. Both HOLMES and our *rewiring* method control pathogen prevalence (marginal of a pathogen incidence over all samples) but only ours control for the inverse marginal of pathogens per stool (number of positives per stool over all pathogens). This difference means that while both methods generally agree on insignificant pairs and agree about four times more than expected at random on significant pairs, there is quite a bit of variability in other pairs found significant. We also note that, with its relaxed assumptions, HOLMES can then easily test for interactions beyond pairs and find only significant interactions in

asymptomatic stools.
(PDF)

**S1 Table. Top 12 pathogen pairs in asymptomatic stools, ranked by their deviation from the ensemble of random graphs, showing the number of co-occurrences and the occurrence as a percent out of asymptomatic stools.**
(PDF)

**S2 Table. Top 12 pathogen pairs in asymptomatic stools, ranked by their deviation from the ensemble of random graphs, showing the number of co-occurrences and the occurrence as a percent out of diarrheal stools.**
(PDF)

## Author Contributions

**Conceptualization:** E. Ross Colgate, Connor Klopfer, Dorothy M. Dickson, Benjamin Lee, Matthew J. Wargo, Beth D. Kirkpatrick, Laurent Hébert-Dufresne.

**Data curation:** E. Ross Colgate, Connor Klopfer, Dorothy M. Dickson, Ashraful Alam.

**Formal analysis:** E. Ross Colgate, Connor Klopfer, Dorothy M. Dickson, Laurent Hébert-Dufresne.

**Funding acquisition:** Beth D. Kirkpatrick.

**Investigation:** E. Ross Colgate, Connor Klopfer, Benjamin Lee, Matthew J. Wargo, Beth D. Kirkpatrick, Laurent Hébert-Dufresne.

**Methodology:** E. Ross Colgate, Connor Klopfer, Dorothy M. Dickson, Laurent Hébert-Dufresne.

**Project administration:** E. Ross Colgate, Matthew J. Wargo, Beth D. Kirkpatrick, Laurent Hébert-Dufresne.

**Resources:** E. Ross Colgate, Ashraful Alam, Beth D. Kirkpatrick, Laurent Hébert-Dufresne.

**Software:** Connor Klopfer.

**Supervision:** E. Ross Colgate, Dorothy M. Dickson, Benjamin Lee, Beth D. Kirkpatrick, Laurent Hébert-Dufresne.

**Validation:** Connor Klopfer, Laurent Hébert-Dufresne.

**Visualization:** E. Ross Colgate, Connor Klopfer, Laurent Hébert-Dufresne.

**Writing – original draft:** E. Ross Colgate, Connor Klopfer, Beth D. Kirkpatrick, Laurent Hébert-Dufresne.

**Writing – review & editing:** E. Ross Colgate, Connor Klopfer, Dorothy M. Dickson, Benjamin Lee, Matthew J. Wargo, Ashraful Alam, Beth D. Kirkpatrick, Laurent Hébert-Dufresne.

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
