## [Decision Letter · Decision Letter 0]

2 May 2023

Dear Dr. Hebert-Dufresne,

Thank you very much for submitting your manuscript "Network analysis of patterns and relevance of enteric pathogen co-infections among infants in a diarrhea-endemic setting" for consideration at PLOS Computational Biology.

As with all papers reviewed by the journal, your manuscript was reviewed by members of the editorial board and by several independent reviewers. In light of the reviews (below this email), we would like to invite the resubmission of a significantly-revised version that takes into account the reviewers' comments.

The age-adjusted expectation of co-occurrence mentioned by Reviewer 2 is potentially very important for these analyses and must be addressed. Further, Reviewer 2 mentions a number of caveats that should be addressed more explicitly in the Discussion (e.g. the absence of pathogen load data as a limitation of how the data were collected; absence of data on potential correlates such as nutrition). In general, care should be taken to report appropriate baselines (e.g. in Figure 5, single-pathogen numbers should be reported). As indicated by Reviewers 1 and 2, the rationale for the network-based approach taken in this manuscript should be explained in the context of alternative analytical approaches, and the structure of the data (in particular, age distribution of samples and distribution of pathogen occurrence as a function of age and sample set) should be made clear.

We cannot make any decision about publication until we have seen the revised manuscript and your response to the reviewers' comments. Your revised manuscript is also likely to be sent to reviewers for further evaluation.

Sincerely,

Nic Vega, Ph.D.

Academic Editor

PLOS Computational Biology

Rob De Boer

Section Editor

PLOS Computational Biology

The age-adjusted expectation of co-occurrence mentioned by Reviewer 2 is potentially very important for these analyses and must be addressed. Further, Reviewer 2 mentions a number of caveats that should be addressed more explicitly in the Discussion (e.g. the absence of pathogen load data as a limitation of how the data were collected; absence of data on potential correlates such as nutrition). In general, care should be taken to report appropriate baselines (e.g. in Figure 5, single-pathogen numbers should be reported). As indicated by Reviewers 1 and 2, the rationale for the network-based approach taken in this manuscript should be explained in the context of alternative analytical approaches, and the structure of the data (in particular, age distribution of samples and distribution of pathogen occurrence as a function of age and sample set) should be made clear.

Reviewer's Responses to Questions

**Comments to the Authors:**

Reviewer #1: Interesting to read this work. Please add and ccompare with some state- of- the art methods with your proposed pipelines

. Also, need to check grammetical typos and errors.

Reviewer #2: This manuscript reports on an analysis of enteric coinfections among infants to identify if specific coinfections occurred more often than would be expected by random chance. This is an important addition to the literature since diarrheal pathogens are often assessed individually despite the ubiquity of coinfections, for which their relevance is unknown. The use of network modeling to maintain pathogen prevalence and pathogen richness is innovative and appropriate. The manuscript is well written.

The primary limitation is that the analysis does not account for the age structure of individual pathogen prevalences. Therefore, it doesn’t account for the fact that coinfections would be expected to be more common for pathogens that have similar age distributions (even if infections with the pathogens are completely independent). For example, Shigella and rotavirus occur together less frequently than expected, but this is could be due to the fact that rotavirus is on average in younger children while Shigella is in older children. Differences in results between the two studies may also be affected by the age distribution of available and analyzed samples since non-diarrheal stools were collected in the first 6 months in PROVIDE when infections were less common.

Second, the analysis of the association between coinfections and diarrhea is limited since it is an ecological analysis (i.e. certain coinfections may be associated with more days of diarrhea overall, but it’s not necessarily the case that the coinfections caused the diarrhea or even that the diarrhea occurred during the coinfections. These associations are likely confounded by host/environmental factors). Since data on pathogens during specific diarrhea episodes is known, this analysis could be done at the level of the diarrhea episode (i.e. are certain co-infections during diarrhea associated with longer duration of that episode). This analysis would also need to be adjusted for confounding factors like age, SES, malnutrition, and other factors associated with children having infections and more severe diarrhea. This might be too much for this manuscript. The novelty of this work is in the network analysis; I would recommend focusing on the primary results that limit the analysis to understanding whether coinfections occur more likely than would be expected by chance, incorporating the age structure of individual pathogen prevalence. Conclusions about which pathogens caused diarrhea (since pathogen quantities are not incorporated) or whether coinfections cause more diarrhea in general are overinterpretations of these analyses.

Specific comments:

1. Author summary (and throughout): suggest toning down causal language: “co-infection increases the number of days with diarrhea”

2. M. tuberculosis is not an enteric pathogen (though it can be detected in stool because children cough and swallow). It would make the most sense to remove that from the analysis. B. fragilis is also not a pathogen (not associated with diarrhea) and I would suggest removing the use of “Enterotoxigenic Bacteroides fragilis.”

3. The rotavirus coinfection results that differ across studies may also be due to age – if vaccination increases the average age of rotavirus infection, it is more likely that vaccinated children will have coinfections with rotavirus and bacteria, since bacteria also more often occur at older ages. It is not clear how these results support the conclusion that rotavirus is likely the causative agent of infection especially in the absence of vaccination. The assumption that the absence of other pathogens means rotavirus is the likely etiology is unstated. This should be moved from the results to the discussion with further explanation. Importantly, the analysis does not incorporate pathogen quantity which is a strong predictor of whether a pathogen present is the etiology or not. This information should be incorporated if conclusions are made about rotavirus being the likely etiology or not (not just whether coinfections were present).

4. Figure S3 reference is missing in the text

5. Figure 5: a simple bar chart might be easier to read here. The comparison in the text of proportion diarrheal for each pathogen pair should be made to the proportion diarrheal for the included individual pathogens, not diarrhea overall. Perhaps this could be incorporated into the figure?

6. Alternatively, the analysis of the proportion of stools for each coinfection that are diarrheal is confounded by the arbitrary ages at which non-diarrheal stool samples were collected (particularly, non-diarrheal stools in PROVIDE only in the first 6 months) and the age distribution of pathogen prevalence. Rather than the proportion diarrheal, a relative risk measure (proportion of diarrheal stools with coinfection / proportion of non-diarrheal stools with coinfection) would account for the different denominators/sampling schemes between diarrheal and non-diarrheal stools

7. The sampling strategy in PROVIDE (mentioned to have been done for a separate analysis) needs to be completely described in the methods. Why were only vaccinated children included from PROVIDE?

8. The last paragraph of the results is an overinterpretation of the data presented due to the concerns described above

9. Figure 7 – pathogens are not defined the same as earlier in the paper (aEPEC and tEPEC separate, for example). Also, there are nodes for both ETEC and LT and ST ETEC, separately? It would be nice to see the results for aEPEC and tEPEC separately throughout the paper since the relevance of aEPEC as a pathogen is unclear.

10. This may be beyond the scope of the paper, but it would be nice to know how generalizable these results are to other sites in MAL-ED

11. Table 1: H. pylori has a typo

12. Fig 2 and 3 legends – there are typos around the bounds > 4 SDs (looks like PDF did not render correctly)

**Have the authors made all data and (if applicable) computational code underlying the findings in their manuscript fully available?**

Reviewer #1: None

Reviewer #2: **No: **I did not see code provided.

PLOS authors have the option to publish the peer review history of their article (what does this mean?). If published, this will include your full peer review and any attached files.

Reviewer #1: No

Reviewer #2: No
---

## [Decision Letter · Decision Letter 1]

5 Sep 2023

Dear Dr. Hebert-Dufresne,

Thank you very much for submitting your manuscript "Network analysis of patterns and relevance of enteric pathogen co-infections among infants in a diarrhea-endemic setting" for consideration at PLOS Computational Biology. As with all papers reviewed by the journal, your manuscript was reviewed by members of the editorial board and by several independent reviewers. The reviewers appreciated the attention to an important topic.

The reviewers indicate that their concerns have been addressed, excepting some minor issues with presentation. Once these issues are addressed, we will be able to accept the manuscript.

Sincerely,

Nic Vega, Ph.D.

Academic Editor

PLOS Computational Biology

Rob De Boer

Section Editor

PLOS Computational Biology

The reviewers indicate that their concerns have been addressed, excepting some minor issues with presentation. Once these issues are addressed, we will be able to accept the manuscript.

Reviewer's Responses to Questions

**Comments to the Authors:**

Reviewer #1: Author's revised where its needed.

Reviewer #2: The authors have appropriately responded to my feedback. I have two remaining comments:

1. I like the addition of Figure 5 and the relative risks. Can you add confidence intervals to the relative risk estimates in the text? It also would be helpful if the RRs are compared to the RRs for the individual pathogens so that the reader can compare the RR for Shigella alone vs. Shigella + rotavirus, for example. I would also recommend adjusting the relative risk estimates for age, or at least acknowledging the potential for confounding by age in these estimates in the discussion.

2. Many of the analyses (including figures) were simply moved to the discussion section and supplement, which is not appropriate (analyses should be in the results section). I maintain my recommendation to remove the diarrhea duration analysis because of the ecological nature (when you have data on individual episodes) and the likely confounding bias. I acknowledge that this was a lot of work, but it detracts from the novelty and impact of the rest of the paper due to the limitations detailed in my last review. Specifically, I would remove starting with "Regarding the association between top pathogen pairs and child health diarrheal outcomes at the individual level..." through "Given the high burden of diarrhea where this pair is present, interventions aimed at disrupting co-infection, as opposed to targeting mono-infection with either pathogen, may reduce the overall burden of enteric disease for infants."

The potential impact on diarrhea outcomes/duration could be mentioned in the discussion section at a higher level without reporting these analyses.

Reviewer #3: The review is attached.

**Have the authors made all data and (if applicable) computational code underlying the findings in their manuscript fully available?**

Reviewer #1: None

Reviewer #2: Yes

Reviewer #3: Yes

PLOS authors have the option to publish the peer review history of their article (what does this mean?). If published, this will include your full peer review and any attached files.

Reviewer #1: No

Reviewer #2: No

Reviewer #3: No

Figure Files:

Data Requirements:

Reproducibility:

References:

---

## [Editor Report · Decision Letter 2]

23 Oct 2023

Dear Dr. Hebert-Dufresne,

We are pleased to inform you that your manuscript 'Network analysis of patterns and relevance of enteric pathogen co-infections among infants in a diarrhea-endemic setting' has been provisionally accepted for publication in PLOS Computational Biology.

Best regards,

Nic Vega, Ph.D.

Academic Editor

PLOS Computational Biology

Rob De Boer

Section Editor

PLOS Computational Biology

---

## [Editor Report · Acceptance letter]

8 Nov 2023

PCOMPBIOL-D-23-00087R2 

Network analysis of patterns and relevance of enteric pathogen co-infections among infants in a diarrhea-endemic setting

Dear Dr Hébert-Dufresne,

I am pleased to inform you that your manuscript has been formally accepted for publication in PLOS Computational Biology. Your manuscript is now with our production department and you will be notified of the publication date in due course.

With kind regards,

Marianna Bach
